Apis mellifera propolis enhances apoptosis and invasion inhibition in head and neck cancer cells

http://orcid.org/0000-0003-2233-918X Niyomtham Nattisa 1
Koontongkaew Sittichai 1
Yingyongnarongkul Boon-ek 2
http://orcid.org/0000-0002-3825-3922 Utispan Kusumawadee 3 kusumawadee.utispan@gmail.com
1 Walailak University International College of Dentistry, Walailak University , Bangkok , Thailand
2 Department of Chemistry, Faculty of Science, Ramkhamhaeng University , Bangkok , Thailand
3 Faculty of Dentistry, Thammasat University , Pathum Thani , Thailand
Uversky Vladimir
Electronic publication date: 2021 Sep 8
Publication date: 2021
Volume: 9
Electronic Location ID: e12139
Received 2021 Jun 3; Accepted 2021 Aug 19
Copyright: © 2021 Niyomtham et al.
Copyright year: 2021
Copyright holder: Niyomtham et al.
License: This is an open access article distributed under the terms of the Creative Commons Attribution License, which permits unrestricted use, distribution, reproduction and adaptation in any medium and for any purpose provided that it is properly attributed. For attribution, the original author(s), title, publication source (PeerJ) and either DOI or URL of the article must be cited.
License URL: https://creativecommons.org/licenses/by/4.0/

Keywords: Head and neck cancer, Apoptosis, Invasion, Propolis

Funding: Faculty of Dentistry, Thammasat University 2561 Center of Excellence for Innovation in Chemistry 2016 This work was supported by the Faculty of Dentistry, Thammasat University (No. 2561) and the Center of Excellence for Innovation in Chemistry (No. 2016). The funders had no role in study design, data collection and analysis, decision to publish, or preparation of the manuscript.

==============================
Background

Propolis is a resinous product accumulated from several plant sources that possess a wide range of therapeutic properties, including anti-cancer activities. However, the role of honeybee-produced propolis on head and neck squamous carcinoma (HNSCC) is not well understood. The aim of this study was to investigate the effects of Apis mellifera propolis on apoptosis and invasiveness in HNSCC cell lines.

Methods

Ethyl acetate extract of propolis (EAEP) was prepared from A. mellifera beehives using liquid–liquid extraction. High-performance liquid chromatography coupled with electrospray ionization-time of flight-mass spectrometry (HPLC-ESI-TOF-MS) was used to determine the flavonoids in EAEP. Isogenic HNSCC cell lines derived from primary (HN30 and HN4) and metastatic site (HN31 and HN12) were used in this study. The cytotoxicity, apoptosis, invasion, and MMP activity of EAEP on HNSCC cells were determined using an MTT assay, flow cytometry, Matrigel invasion assay, and gelatinase zymography, respectively.

Results

We found that EAEP exhibited cytotoxic activity and induced apoptosis in the HNSCC cell lines. Furthermore, EAEP significantly decreased HNSCC cell invasion by reducing MMP-2 and MMP-9 activity. Two flavonoids, galangin and apigenin, were identified in EAEP by HPLC-ESI-TOF-MS. The results suggest that EAEP promotes apoptosis and exerts anti-invasion potential by inhibiting MMP-2 and MMP-9 activity in HNSCC cell lines. These inhibitory effects may be mediated by galangin and apigenin.

Introduction

Head and neck squamous cell carcinoma (HNSCC) is the seventh most common cancer worldwide and the highest occurring cancer observed in southern Asia. Currently, the typical HNSCC treatment is surgical removal, combined with chemotherapy and/or radiation therapy (Chow, 2020; Schwartz & Hayes, 2020). Similar to other tumors, proliferation, invasion and metastasis are the critical processes that indicate HNSCC aggressiveness (Chan et al., 2016; Wolf & Claudio, 2014). Evading apoptosis is one way by which cancer cells survive in an extreme microenvironment (Raudenska, Balvan & Masarik, 2021). HNSCC invasion and metastasis are driven by matrix metalloproteinase (MMP) activity. MMP-2 and MMP-9 are the key enzymes that destroy the basement membrane and degrade the extracellular matrix, leading to tumor invasion (Koontongkaew, 2013). Thus, more effective treatments that trigger apoptosis in cancer cells for local and metastatic HNSCC are needed (Khan et al., 2012).

Propolis, or bee glue, is a natural resinous material collected by honey bees from various tree buds to seal cracks in the hive and protect the hive against bacterial and fungal infection (Calegari et al., 2017). Propolis has been used in traditional medicine in many countries. More than 300 chemical compounds have been identified from propolis in different geographic regions (Drescher et al., 2019; Xuan et al., 2016), including flavonoids, terpenes, phenolic acid, cinnamic acid, caffeic acid, and several esters (Funakoshi-Tago et al., 2016; Jaiswal et al., 1997; Kocot et al., 2018). Propolis has a wide range of pharmaceutical properties, including antimicrobial (Al-Ani et al., 2018; Chen et al., 2018), anti-inflammatory, antioxidant (Kocot et al., 2018), anti-angiogenic (Iqbal et al., 2019) and anti-cancer (Badr et al., 2011; Frozza et al., 2017; Sawicka et al., 2012) effects. The crude extracts of propolis have demonstrated cytotoxic activity against various cancer cell lines, such as human prostate cancer cells (DU145 and PC-3) (Li et al., 2007), cervix adenocarcinoma cells (HeLa) (Barbaric et al., 2011), human laryngeal epidermoid carcinoma cells (Hep-2) (Frozza et al., 2017), human colorectal adenocarcinoma cells (HT-29), human breast adenocarcinoma cells (MCF-7), human epithelial colorectal adenocarcinoma cells (Caco-2) and murine melanoma cell lines (B16F1) (Choudhari et al., 2013).

It has been reported that the biological and pharmacological activities of propolis depend on its chemical composition, geographical zone, plant source, and season (Devequi-Nunes et al., 2018; Omar et al., 2017; Siheri et al., 2016). Propolis extracts from Apis mellifera beehives in Thailand have demonstrated anti-proliferative and cytotoxic activity against cancer cell lines derived from human breast carcinoma (BT474), human hepatocellular carcinoma (Hep-G2), gastric carcinoma (KATO-III), and colon adenocarcinoma (SW620) (Teerasripreecha et al., 2012). Moreover, the propolis extract from Trigona sirindhornae exhibited cytotoxic effects against HNSCC cells (Utispan, Chitkul & Koontongkaew, 2017). However, there are few studies on the effect of Thai A. mellifera propolis extract on HNSCC cell lines. Therefore, the aim of this study was to investigate the cytotoxic, apoptotic, and anti-invasive activity of the ethyl acetate extract from Thai A. mellifera propolis on primary and metastatic HNSCC cell lines.

Materials and Methods

Chemicals

Apigenin, galangin, caffeic acid, ferulic acid, rutin, quercetin, and naringenin were purchased from Sigma-Aldrich (St. Louis, MO, USA). Acetonitrile (HPLC grade) was purchased from RCI Labscan (Bangkok, Thailand). Hexane, ethyl acetate, ethanol, methanol, and formic acid (analytical grade) were purchased from Merck (Darmstadt, Germany).

Ethanol extract of propolis (EEP) preparation

The propolis sample from the native Thai bee species A. mellifera was obtained in November 2017 in Loei province, northeastern Thailand. The sample was stored in a desiccator and kept in the dark at 4 °C until processed. Raw propolis (5.27 g) was cut into small pieces and stirred in 100 ml 95% (v/v) ethanol (EtOH) at 100 rpm at room temperature for 48 h in the dark. The insoluble portion was separated by filtration through No. 2 Whatman filter paper (Whatman Inc, Piscataway, NJ, USA). To increase the extract yield, this procedure was repeated three times on the same sample. The resulting filtrates were pooled and dried in a rotatory evaporator at 40 °C and 175 mbar (Rotovapor R-215, BUCHI Labortechnik, AG, Switzerland). The ethanol extract of propolis (EEP, 4.32 g) with a viscous appearance was obtained.

Liquid–liquid partitioning

EEP was fractioned using liquid-liquid partitioning. The EEP (4.32 g) was dissolved in 100 ml methanol and then partitioned with hexane (3 × 50 ml). The combined hexane extract was then rotatory evaporated at 40 °C and 335 mbar to yield the hexane extract of propolis (0.98 mg). The methanol portion was evaporated at 40 °C and 337 mbar. The methanol extract was dissolved in 100 ml distilled water and underwent liquid-liquid partitioning with ethyl acetate (3 × 50 ml). After the ethyl acetate was evaporated under reduced pressure on a rotary evaporator, the residual solvent was removed by drying under vacuum (Edwards RV12 rotary vane vacuum pump, Bolton, England) at room temperature for 16 h. The residual ethyl acetate in ethyl acetate extract of propolis (EAEP) was determined by Nuclear Magnetic Resonance (NMR) Spectroscopy. The crude extract was dissolved in DMSO-d6 (Sigma-Aldrich, St. Louis, MO, USA) and the 1H-NMR spectrum was recorded on a Bruker ASCEND 400 FT-NMR spectrometer (Bruker, Faellanden, Switzerland) operating at 400 MHz. The chemical shifts are shown as parts per million (ppm). Finally, solid masses were obtained for the ethyl acetate (1.92 g) and aqueous (0.12 g) extracts of propolis after complete solvent evaporation.

Most of the substances found in propolis were obtained in polar organic solvents, including ethanol, methanol, and ethyl acetate (Sambou et al., 2020). Solvents, e.g. ethyl acetate, are used in extraction processes because of its chemical and biological functions, such as its medium polarity (polarity index = 4.3) (Synder, 1974). The biphasic action of this solvent enables it to be used to extract both polar and non-polar compounds (Mandal, Mandal & Das, 2015). Ethyl acetate is classified as a class 3 solvent with low toxic potential (Abarca-Vargas, Pena Malacara & Petricevich, 2016; Aru et al., 2019). Furthermore, previous studies demonstrated that the ethyl acetate extracts of many medicinal plants were not cytotoxic to normal cells (El Khalki et al., 2018; Jain et al., 2011; Seklic et al., 2018). Therefore, only EAEP was used in the present study. Before use, the EAEP extract was dissolved in dimethyl sulfoxide (DMSO) and placed in a freezer (−30 °C) until use.

Cell culture

Two pairs of isogenetic HNSCC cell lines representing primary and metastatic disease from the same patient were first established at Wayne State University by Ensley J. (Cardinali et al., 1995) who collaborated with a researcher at the National Institute of Dental and Craniofacial Research under the supervision of Gutkind S. Gutkind S. provided the cell lines as a gift to Koontongkaew S. The HN30 and HN31 cells were obtained from primary pharynx lesions and lymph node metastases (T3N1M0), respectively. The HN4 and HN12 cells were obtained from primary tongue lesions and lymph node metastases (T4N1M0), respectively. The cells were maintained in Dulbecco’s Modified Eagle’s Medium (DMEM) (Invitrogen, Carlsbad, CA, USA) supplemented with 10% fetal bovine serum, 100 U/ml penicillin, 100 μg/ml streptomycin (Invitrogen) and an anti-fungal agent. The cells were cultured in a 37°C humidified 5% CO2 atmosphere. The cells were passaged with 0.25% trypsin-EDTA when reaching 70–80% confluence.

Cell viability-MTT assay

The cytotoxicity of EAEP on the HNSCC cell lines was investigated using the methyl thiazotetrazolium (MTT) as previously described (Utispan et al., 2020). Basically, the incubation periods for the MTT assay were 24, 48, or 72 h. However, tumor cells are very heterogenous and their doubling times vary. Therefore, a 24 h incubation period was enough if the doubling time was ~16–24 h. However, for certain cancer cell lines, a 24 h incubation period with a given test substance is too short to demonstrate a significant effect on cell viability (Abdul Latif et al., 2019; Alizadeh-Navaei et al., 2016; Guo et al., 2015). Therefore, in the present study, a 72 h incubation period was used to assess cell toxicity. The HNSCC cells were seeded in 96-well plates at a density of 2,000 cells/well. The cells were treated with serum-free DMEM with 0.1% DMSO (vehicle control) or EAEP (0.10–0.40 mg/ml) at 37 °C for 72 h. After the exposure period, the media was removed, and the cells were washed with phosphate-buffered saline (PBS), and incubated with 0.5 mg/ml MTT (Sigma) in culture media for 4 h. The purple formazan crystals of the viable cells were dissolved and measured at 570 nm by a microplate reader (Tecan, Salzburg, Austria). Cell viability was calculated as a percentage of that of the control (untreated) cells. Each EAEP concentration was independently assayed three times with three technical replicates. Based on ISO 10993-5, cell viability above 80% was considered non-cytotoxic, between 80–60% weak, 60–40% moderate, and below 40% strong cytotoxicity, respectively (International Organization for Standardization ISO 10993-5, 2009).

Apoptosis assay

To verify that the effect of the studied extracts on the growth inhibition of HNSCC cells was related to apoptosis, the apoptosis and necrotic cells were analyzed using annexin-V- fluorescein isothiocyanate (FITC) and propidium iodide (PI) staining. The cells were seeded in 6-well plates and allowed to attach for 24 h. The cells were then treated with EAEP at the weak cytotoxic dose (cell viability of 60–80%) for 24 h. The HN12, HN30 and HN31 cells were treated with 0.2 mg/ml EAEP, whereas the HN4 cells were treated with 0.3 mg/ml EAEP. The cells were washed twice with PBS and detached by 0.25% trypsin, washed with PBS and resuspended in ice cold binding buffer. The apoptotic cells were assessed using the BD Annexin V FITC Assay (BD Biosciences, San Jose, CA, USA). Ten thousand events were analyzed in a flow cytometer (Cytoflex®, Beckman Coulter, Indianapolis, IN, USA). The percent viable, apoptotic and necrotic cells were determined by CytExpert Software (Beckman Coulter).

Invasion assay

The modified Boyden chemotaxis chamber (Neuro Probe, Gaithersburg, MD, USA) assay used for the cell invasion analysis is based on a chamber with two medium filled compartments as previously described (Albini et al., 1987). Matrigel, a reconstituted basement membrane gel (BD Bioscience) was applied to a polycarbonate membrane filter (13 mm-diameter, 8.0 μm pore size, Whatman). The filter was placed above the lower chamber that contained serum-free DMEM with 0.1% bovine serum albumin (BSA; Sigma). HN4, HN12, HN30, or HN31 cells (1 × 105 cells) were resuspended in 0.1 mg/ml EAEP diluted in DMEM containing 0.1% BSA and seeded into the upper well of the chamber. The cells were incubated for 5 h, then the filters were fixed and stained with crystal violet for 10 min. The invaded cells were counted by two investigators using a microscope at 400×g magnification. The invaded cell counts were averaged from five randomly selected fields.

Conditioned medium preparation and zymography

HN4, HN12, HN30, or HN31 cells (1 × 106 cells) were cultured in 6-well plates and incubated at 37 °C for 24 h. After incubation, the wells were washed with PBS and treated with 0.1 mg/ml EAEP diluted in DMEM containing 0.1% BSA for 48 h. Cells cultured in DMEM containing 0.1% BSA were used as control. The conditioned medium (CM) was collected as previously described in Utispan et al. (2020).

The MMP-2 and MMP-9 activity in the CM were measured using gelatin zymography as previously described (Koontongkaew, Amornphimoltham & Yapong, 2009). The clear bands in the gel were identified as degrading activity of MMPs. Images of the stained gels and the gelatinolytic band quantification were processed using the G:BOX gel documentation system (Syngene, Frederick, MD, USA) and Gene Tools software (Syngene), respectively. Three independent experiments were performed.

HPLC-ESI-TOF-MS analysis of EAEP

High-performance liquid chromatography coupled with electrospray ionization-time of flight-mass spectrometry (HPLC-ESI-TOF-MS) was used to investigate selected phenolic acids and flavonoids in EAEP. EAEP was prepared at 5 mg/ml in ethanol and filtered through a 0.45 μm membrane filter. Although a large number of natural compounds have been reported in propolis, we focused on specific substances in propolis that exhibit anti-cancer activity (Anjum et al., 2019), Thus, we investigated the presence of apigenin, galangin, caffeic acid, ferulic acid, rutin, quercetin and naringenin in EAEP. To identify the compounds, seven standard polyphenolic compounds (apigenin, galangin, caffeic acid, ferulic acid, rutin, quercetin and naringenin) were dissolved in methanol (10 ppm). The analyses were conducted in an UltiMate® 3000 system (Thermo Fisher Scientific, DionexSoftron GmbH, Dornierstr. 4, Germany) with a reverse phase column (C18 analysis column, 2.1 mm × 150 mm and 3 µm particle size, Thermo Fisher Scientific, Sunnyvale, CA, USA) at 40 °C. The injection volume for all samples was 5 μl. The mobile phase consisted of solvent (A) 0.1% (v/v) formic acid in water, and solvent (B) acetonitrile, which had been degassed and filtered. The gradient program for the HPLC was: 0–1 min, 5% B; 1–40 min 5–55% B; 41–50 min 55–95% B; 50.1–55 min 5% B, and the flow rate was 0.3 ml/min.

The separated components from the HPLC system were subjected to mass to charge ratio (m/z) analysis using an ESI-TOF-MS system. ESI-TOF-MS was carried out using a time of flight mass spectrometer (micrOTOF-Q-II, Bruker Daltonik, Bremen, Germany). An electrostray ion source (ESI) was used in negative ion mode (ESI-) with the following settings: capillary voltage 3 kV, nebulizer gas pressure 2.0 bar, dry gas temperature 200 °C, and dry gas flow rate 8.0 L/min. Spectra were collected from m/z 50–1,000 Da. Bruker Compass Data Analysis 4.0 software (Bruker Daltonik, Billerica, MA, USA) was used for recording and processing the data. The phenolic and flavonoid content in EAEP was determined by interpolation with a calibration curve constructed using the standard solutions of selected polyphenols. The determination was performed three times.

Statistical analysis

Data analyses were performed using GraphPad Prism 7.04 software (GraphPad Software, La Jolla, CA, USA). All results are expressed as means and standard error of the mean (SEM) values from three independent experiments. P ≤ 0.05 was considered statistically significant using one-way ANOVA followed by Dunnett’s multiple comparison.

Results

Cytotoxic assessment of EAEP on HNSCC cell lines

The cytotoxic effect of EAEP was evaluated using an MTT assay. The EAEP (0.2 mg/ml) significantly decreased the HN30 and HN12 cell viability compared with the control in a dose-dependent manner, whereas the HN4 and HN31 cells demonstrated significantly reduced cell viability at 0.25 and 0.1 mg/ml EAEP, respectively in a dose-dependent manner (Figs. 1A–1D). The IC50 value of EAEP for the HN30, HN31, HN4 and HN12 cells were 0.19, 0.16, 0.31, and 0.14 mg/ml, respectively. However, according to ISO 10993-5, the non-cytotoxic dose (cell viability above 80%) for the HN12, HN30 and HN31 cells was 0.1 mg/ml EAEP, whereas 0.2 mg/ml EAEP was not toxic to the HN4 cells. Therefore, the non-cytotoxic concentration at 0.1 mg/ml was selected to treat the HNSCC cell lines and evaluate their invasion and MMP activity in the subsequent experiments. The weak cytotoxic dose (cell viability of 60–80%) was used for the apoptosis assay.

Figure 1 Cytotoxic evaluation of EAEP on HNSCC cells measured by MTT assay.

The EAEP at various concentrations were used to treat (A) HN30, (B) HN31, (C) HN4 and (D) HN12 cells for 72 h. Bars represent means ± SEM of three independent experiments (n = 3). *P < 0.05 compared with the control.

Apoptotic effects of EAEP

Based on the effect of EAEP on HNSCC cell viability, we hypothesized that apoptosis was an underlying mechanism. Flow cytometry was used to quantify the apoptotic, alive, and necrotic cells. The HN12, HN30, and HN31 cells were exposed to 0.2 mg/ml EAEP. The HN4 cells were treated with 0.3 mg/ml EAEP. We found that EAEP induced 42.42%, 44.01%, 43.28% and 13.08% apoptosis in the HN30, HN31, HN4, and HN12, cells respectively (Fig. 2).

Figure 2 Effect of EAEP on HNSCC cells, after treatment for 24 h, evaluated by flow cytometry.

(A) Dot plots of apoptosis assay with Annexin V-FITC (X-axis)/ 7-AAD-PE (Y-axis). (B) Percentage of apoptotic HN30, HN31, HN4, and HN12 cells treated with 0.2, 0.2, 0.3 and 0.2 mg/ml EAEP, respectively for 24 h. Bars represent means ± SEM of three independent experiments (n = 3). *P < 0.05 compared with the control.

EAEP decreased HNSCC invasion

HNSCC cell invasion using a Boyden chamber system was used as a functional assay. We found that HNSCC invasion was down-regulated after EAEP treatment for 5 h. EAEP at the non-toxic concentration (0.1 mg/ml) decreased the invasiveness of the HN30, HN31, HN4 and HN12 cells by approximately 51%, 67%, 37% and 56%, respectively, compared with control (Figs. 3A and 3B).

Figure 3 EAEP decreased HNSCC cell invasion.

A non-cytotoxic dose of EAEP was used to treat HN30, HN31, HN4 and HN12 cells. (A) Representative images of cells that invaded onto the underside of polycarbonate filters coated with Matrigel membrane from Boyden chemoinvasion assays at 400×g magnification using a light microscope. (B) The number of cells of each field was counted using a microscope at 400×g magnification. Values are the mean ± SEM of three independent experiments (n = 3) *P < 0.05 compared with the control.

EAEP reduced the MMP activity of HNSCC cells

To determine whether MMPs were down-regulated in EAEP-treated HNSCC cells, the cell culture medium of the HNSCC cells was assayed for MMP-2 and MMP-9 activity after treatment with EAEP for 48 h. The zymographic data demonstrated that EAEP inhibited the MMP-2 and MMP-9 activity of the HN30, HN31, HN4, and HN12 cells (Fig. 4A). Quantitative analysis of the MMP activity revealed that at the non-toxic concentration (0.1 mg/ml), EAEP significantly reduced the MMP-2 activity of the HN30 and HN31 cells by 64% and 27%, respectively (Fig. 4B). However, the MMP-2 activity in the EAEP-treated HN4 and HN12 and control cells was not significantly different. In contrast, 0.1 mg/ml EAEP significantly decreased MMP-9 activity in the HN30, HN31, HN4 and HN12 cells to approximately 51%, 32%, 18% and 21%, respectively.

Figure 4 EAEP reduced MMP activities.

EAEP (0.1 mg/ml) was used to treat the cancer cells for 48 h and MMP activity in the conditioned media of (A) HN30 and HN31, and (B) HN4 and HN12 cells were detected using zymography. GeneTools software was used to quantify gelatinolytic bands of (C) MMP-2 and (D) MMP-9 activity. Bars represent means ± SEM of three independent experiments (n = 3). *P <0.05 compared with the control.

HPLC-ESI-TOF-MS analysis of EAEP

HPLC-ESI-TOF-MS was used to analyze the profiles of the phenolics and flavonoids in EAEP. Commercially available polyphenolic compounds were used as standards in this determination. Baseline calibration of the HPLC system was performed using the sample solvent (Fig. 5A). Although minor peaks were present, only two compounds were detected in a measurable quantity. The EAEP chromatograms demonstrated peaks 1 and 2 with retention times that corresponded to those of apigenin (24.7 min) and galangin (32.9 min), respectively (Figs. 5B and 5C). The HPLC-ESI-TOF-MS parameters were optimized and used to profile EAEP. The selected 2 compounds in EAEP were putatively identified by comparison to the database (Table 1). The results revealed that compounds 1 and 2 were apigenin and galangin, respectively. The amount and chemical structures of apigenin and galangin are shown in Fig. 6. The concentration of these phenolic compounds in EAEP was estimated by interpolation with a calibration curve constructed with standard solutions of apigenin and galangin. The quantitative determination revealed that the amount (mean ± SD) of apigenin and galangin in the EAEP was 149.0 ± 7.07 µg/g and 628.66 ± 16.42 µg/g, respectively.

Figure 5 HPLC-ESI-TOF-MS analysis of EAEP.

(A) Total ion chromatogram (TIC) of blank solution, (B) EAEP sample, and (C) standard compounds (apigenin (Cmpd 1, 24.7 min) and galangin (Cmpd 2, 32.9 min)) by negative mode HPLC-ESI-TOF-MS.

Figure 6 Concentration and putative structure of compounds in EAEP.

(A) Concentrations of apigenin and galangin in EAEP measured by HPLC-ESI-TOF-MS. (B) The chemical structure of apigenin and galangin.

Table 1 Retention time, calculated and detected masses, calculated formula, concentration and putative identification of the two compounds in EAEP analyzed by HPLC-ESI-TOF-MS.

Peak	Retention time
(min)	Calculated mass (M-H)– (m/z)	Detected mass (M-H)– (m/z)	Calculated formula (M-H)–	Concentration (µg/g)	Putative identification	
Cmpd 1	24.8	269.0455	269.0465	C15H9O5	149.0	Apigenin	
Cmpd 2	33.0	269.0455	269.0471	C15H9O5	628.6	Galangin	

Detection of residual solvent in EAEP

The final product extract should not contain any residual solvent to ensure that the biological activities observed were not due to ethyl acetate in the test fraction. The residual solvents were evaporated under reduced pressure on a rotatory evaporator, followed by residual solvent removal in a vacuum dryer for 16 h. Further, 1H-NMR analysis of the ethyl acetate confirmed the absence of residual ethyl acetate in the EAEP. Three sets of protons were responsible for the three signals in the 1H-NMR spectra of ethyl acetate (Fig. 7A). The two Hb protons in ethyl acetate split the Hc signal into a triplet at approximately δ 1.70 ppm, and the three Hc protons split the Hb signal into a quartet at approximately δ 4.02 ppm. There was an unsplit single peak at approximately δ 1.98 ppm that corresponds to the acetyl (Ha) protons. The 1H-NMR spectra were consistent with a previous report (Fulmer et al., 2010). The 1H-NMR spectrum of EAEP did not exhibit signals identifying ethyl acetate (Fig. 7B). Therefore, the 1H-NMR spectra indicated that the EAEP extract was not contaminated with ethyl acetate. This confirms that the bioactive compounds in EAEP were responsible for the cytotoxicity, apoptosis, cell invasion, and MMP activity of HNSCC cell lines in our studies.

Figure 7 NMR spectrum of ethyl acetate.

(A) 1H-NMR spectra (400 MHz) of ethyl acetate in DMSO-d6. Three sets of protons are responsible for the three signals in the 1H-NMR spectra of ethyl acetate. The signals at approximately δ 1.70, 1.98 and 4.02 ppm correspond to the Ha, Hb and Hc hydrogens, respectively. (B) The 1H-NMR spectra (400 MHz) of EAEP in DMSO-d6 did not exhibit signals identifying ethyl acetate.

Discussion

The main objective of the present study was to evaluate the anti-cancer effect and establish the underlying mechanisms of Thai propolis on HNSCC cells. This study investigated, for the first time, the anti-cancer effect of the ethyl acetate extract of propolis (EAEP) from Thai A. mellifera on primary and metastatic HNSCC cell lines. We found that EAEP demonstrated a dose-dependent cytotoxic effect and caused apoptosis in the HN30, HN31, HN4 and HN12 cell lines. Our findings agree with previous studies demonstrating the anti-cancer effects of propolis obtained from many countries. Brazilian propolis extract inhibits cell growth and induces apoptotic mechanisms in human prostate carcinoma (DU145 and PC-3 cells) (Li et al., 2007). Similarly, a propolis extract from Turkey inhibited cell proliferation and induced apoptosis and cell cycle arrest in breast cancer (MCF7), lung cancer (A549) and gastric cancer (HGC27) (Aru et al., 2019; Misir et al., 2020).

We analyzed standards of apigenin, galangin, caffeic acid, ferulic acid, rutin, quercetin and naringenin, however; only apigenin and galangin were observed in our propolis samples. This finding demonstrated that propolis extracts differ qualitatively and quantitatively regarding phenolic acids and flavonoids (Anjum et al., 2019). Apigenin (Swanson et al., 2014; Yan et al., 2017; Zhu et al., 2016) and galangin (Seyhan et al., 2019; Yang et al., 2018; Zhu et al., 2014) have demonstrated anti-cancer activity against various cancer cell lines and they might be involved in the cytotoxic effect on HNSCC cell lines. These results suggest that the cytotoxic activities of phenolic compounds depends on their chemical structure, especially the total number of hydroxyl groups in their molecules (Czyzewska et al., 2016). Here, we demonstrated the inhibition of HNSCC cell proliferation by propolis extracts could occur, at least partially, through apoptosis. Apigenin and galangin may play an important role in the EAEP cytotoxic effect on HNSCC cell lines. The mechanism by which EAEP caused apoptosis in HNSCC cells is unclear. These phenolic compounds may induce apoptosis in HNSCC cells by decreasing the expression of anti-apoptotic proteins and increasing the expression of pro-apoptotic proteins (Zhu et al., 2014). EAEP might modulate the caspase-3 and AKT signaling pathways (Wang & Tang, 2017). However, the synergistic effects of polyphenols in the propolis extract might be responsible for their cytotoxicity (Czyzewska et al., 2016).

Invasion and migration are considered important hallmarks of malignant tumors. MMP-2 and MMP-9 are enzymes that play an important role in basement membrane degradation, which is the first step in the invasion and metastasis of cancer cells (Koontongkaew, 2013). In the present study, we investigated the effects of EAEP on cell invasion by focusing on the activity of MMP-2 and MMP-9 in HNSCC cell lines. Notably, EAEP significantly decreased the invasion of stage III, HN30 and HN31 cells by inhibiting MMP-2 and MMP-9 activity. However, the extract reduced invasion of stage IV, HN4 and HN12, cells by attenuating only MMP-9 activity. Previous studies found that apigenin inhibited the invasion and migration of human metastatic cancer cell lines by reducing MMP-9 expression by suppressing the p38 MAPK signaling pathways (Noh et al., 2010). Moreover, galangin reduces MMP-9 expression and cell migration in human neuroblastoma cell lines (Yang et al., 2018) and human fibrosarcoma cells (Choi, Lee & Lee, 2015). Based on these and our findings, we hypothesize that apigenin and galangin in EAEP may be a key factor in inhibiting invasive HNSCC cells.

Previous studies have reported the cytotoxic and anti-invasiveness concentrations of apigenin and galangin in the range 0.01–0.1 and 5–30 µg/ml, respectively (Seyhan et al., 2019; Swanson et al., 2014). In the present study, we obtained 1,920 mg EAEP and found approximately 286.08 and 1,207 µg apigenin and galangin, respectively, in the extract. We estimated that the effective dose of EAEP contains maximum concentrations of 0.045 and 0.188 µg/ml of apigenin and galangin respectively. These results imply that apigenin and galangin are at least two candidate compounds that exhibit anticancer effects on HNSCC cells. Based on the results of the present study, other compounds in EAEP should be identified and their synergistic effect investigated.

One limitation of our study is that we did not compare the biological activities of EAEP between HNSCC cells and normal cells. Therefore, it is difficult to specify that the EAEP activities are specific to cancer cells. To clarify these observations, further studies are needed to investigate the anti-cancer activities in counterpart normal cells. Additional in vivo studies should be undertaken to determine the efficacy of EAEP and its related compounds in inhibiting tumor progression.

Conclusions

In conclusion, the present study revealed that EAEP from Thai A. mellifera has a dose-dependent cytotoxic effect and induces apoptosis of HNSCC. The EAEP inhibited the invasion of primary and metastatic HNSCC cells by inhibiting MMP-2 and MMP-9 expression. Apigenin and galangin were identified in the EAEP. These two flavonoids may contribute to the anti-cancer activities of EAEP. Therefore, the EAEP has the potential to be a powerful candidate in developing preventive agents for cancer metastasis and this beneficial effect may expand future research on the anticancer properties of EAEP in vitro and in vivo.

Supplemental Information

Supplemental Information 1 HPLC-ESI-TOF-MS setting of blank.

Click here for additional data file.

Supplemental Information 2 HPLC-ESI-TOF-MS setting of EAEP.

Click here for additional data file.

Supplemental Information 3 HPLC-ESI-TOF-MS setting of standard compounds.

Click here for additional data file.

Supplemental Information 4 Statistical analysis of numerical data.

Click here for additional data file.

Supplemental Information 5 Uncropped gels.

Zymographic gels for indicating MMPs activity in HN30, HN31, HN4 and HN12 cell lines.

Click here for additional data file.

Supplemental Information 6 Raw data.

The raw data provide numerical data of MTT assay, apoptosis, invasion assay and MMPs activity.

Click here for additional data file.

The authors thank Professor Silvio Gutkind (Moores Cancer Center, Department of Pharmacology, UCSD, CA, USA) for the HNSCC cell lines used in our study. We thank Christian Estacio for his assistance in editing the English of this manuscript.

Additional Information and Declarations

Competing Interests

Author Contributions

Data Availability

The authors declare that they have no competing interests.

Nattisa Niyomtham performed the experiments, analyzed the data, prepared figures and/or tables, authored or reviewed drafts of the paper, and approved the final draft.

Sittichai Koontongkaew analyzed the data, prepared figures and/or tables, authored or reviewed drafts of the paper, and approved the final draft.

Boon-ek Yingyongnarongkul analyzed the data, prepared figures and/or tables, authored or reviewed drafts of the paper, and approved the final draft.

Kusumawadee Utispan conceived and designed the experiments, performed the experiments, analyzed the data, prepared figures and/or tables, authored or reviewed drafts of the paper, and approved the final draft.

The following information was supplied regarding data availability:

The raw data of HPLC-ESI-TOF-MS condition and numerical data with statistical analysis are available as Supplemental Files.

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
