# Peer review of "Apis mellifera propolis enhances apoptosis and invasion inhibition in head and neck cancer cells"

_PeerJ, doi:10.7717/peerj.12139_

## Round 0.1 · original submission · Major Revisions

Please address all the critiques of both reviewers and revise the manuscript accordingly.

Reviewer 1 ·

Basic reporting

The manuscript has some originality, technical quality, clarity of production and it is important in its field.
The “introduction” has background, describe a problem, shows what currently exists and proposes solution, with current literature references.
A revision of the English language can improve the quality of the work.

Experimental design

The methodology is well described and reproducible.

Validity of the findings

Conclusions are well stated, linked to the original research question & limited to supporting results.

Additional comments

Major points: The strongest point of the work was the use of two pairs of isogenetic HNSCC cell lines representing both primary and metastatic disease from the same patient.

The use of doses with low cytotoxicity can somehow compensate for the lack of use of non-tumor cells, which would assess the selectivity of treatment for tumor cells.
Also, the evaluation of other secondary compounds that might be present in the EAEP should have been done.
The work's weakest point was the lack of any in vivo testing. It was written in an understandable language, but it still needs language revision to improve clarity.

Reviewer 2 ·

Basic reporting

The paper is interesting and provides information about anti tumor activity of a especific fraction of propolis from Tai Apis mellifera.
However, this revisor would like to understand the reasons why only EAEP was studied and not the first extract obtained, and either, why not galangin and apigenin as isolated compounds as the authors are attributing the activities found in these two substances.

Experimental design

About the experimental design, I have some points to the authors answer:

1. I would like to understand the reasons why the authors used 72 hours to measure citotoxicity. Isn't it to high for cell culture citotoxicity evaluation?

2. The authors used 24 hours for flow citometry measure, 48 hours for MMP activity determination and 72 hours for cell viability. Please, justify the reasons.

3. The authors atribute the anti tumor activities to galangin and apigenin. However, the EAEP fingerprint observed in the figure 5 b, showed other important peaks around the minutes 43-45. Galangin and apigenin are easily available in the market and the authors used them to construct the calibration curve. Why the author didn't used the isolated compound to test their hypothesis?

4. Considering that the authors possess an HPLC-ESI-TOF-MS, why don't identify the other peaks present in the EAEP? Especially considering that other compounds maybe be present in amounts closer to galangin and apigenin in the fraction (figure 5b), and probably can contribute the the findings, it is important to identify then.

5. Considering the solvents used in the extraction process, can the authors provide evidences about the lack of solvents residues in the samples tested? Or, the controls involved the presence of these solvents?

6. In the lines 142-143 the authors inform that ethyl acetate is used as a polar organic solvent. In the lines 143-145, the authors says that is a medium polarity solvent and with minimal toxicity. Can the authors inform the reference for both information? Especially the minimum cell toxicity?

7. Considering that the authors atribute to the EAEP powerful citotoxicity in tumor cells studied, can the authors support that this EAEP extract is safety to the normal cells? Is is very important to check the citotoxicy in at least fibroblast cells in order to attribute to this EAEP extract a powerful candidate for future in vivo studies.

Validity of the findings

In the conclusions, the author attribute to the extract obtained a powerful candidate to anti tumor activities. Also, the authors discussed that other authors considered galangin and apigenin this activity. Can the authors discuss the concentration of these substances in the EAEP obtained, the scientific literature already published with these compounds to the same activity, and the dosages found as effective in the paper to support "powerful candidate" term in the conclusion ?

---

## Round 0.2 · accepted · Accept

Thank you for addressing the concerns of the reviewers. I am pleased to accept your revised manuscript in its current form.

Reviewer 1 ·

Basic reporting

The authors reviewed, edited, and answered all questions in a satisfactory manner.
Reviewers' comments helped the authors improve the text, and the manuscript was revised by a native speaker. I think that it deserves to be published.

Experimental design

The experimental design was revised.

Validity of the findings

The modifications met the requested criteria. Information added in the discussion item clarified the text.

Annotated reviews are not available for download in order to protect the identity of reviewers who chose to remain anonymous.

Reviewer 2 ·

Basic reporting

The manuscript was improved with the points previously requested. The manuscript is clear, with professional structure, good background and english.

Experimental design

The questions previously done were properly answered.

Validity of the findings

The limitations was identified clearly in the manuscript.